# Impact of the COVID-19 Pandemic on the Use of Antidepressants by Young Adults in the ASL TO4 Regione Piemonte (Italy)

**DOI:** 10.3390/pharmacy12010021

**Published:** 2024-01-23

**Authors:** Lucrezia Greta Armando, Raffaella Baroetto Parisi, Cristina Rolando, Mariangela Esiliato, Valeria Vinciguerra, Cecilia Bertiond, Abdoulaye Diarassouba, Clara Cena, Gianluca Miglio

**Affiliations:** 1Department of Drug Science and Technology, University of Turin, Via Pietro Giuria 9, 10125 Turin, Italy; lucrezia.armando@edu.unito.it; 2Struttura Complessa Farmacia Territoriale ASL TO4, Regione Piemonte, Via Po 11, 10034 Chivasso, Italy; rbaroettoparisi@aslto4.piemonte.it (R.B.P.); crolando@aslto4.piemonte.it (C.R.); mesiliato@aslto4.piemonte.it (M.E.); vvinciguerra@aslto4.piemonte.it (V.V.); cbertiond@aslto4.piemonte.it (C.B.); adiarassouba@aslto4.piemonte.it (A.D.); 3Competence Centre for Scientific Computing, University of Turin, Corso Svizzera 185, 10149 Turin, Italy

**Keywords:** antidepressant use, mental health status, young adults, COVID-19 lockdown, ARIMA

## Abstract

The association between younger age and poorer mental health during the COVID-19 pandemic has been documented. Whether these changes were associated with a change in antidepressant (AD) use is not well understood. This study aimed to estimate the impact of the COVID-19 pandemic on AD use by young adults in the ASL TO4 Regione Piemonte (Italy). The impact of the pandemic on the weekly prevalence of AD users was assessed using interrupted time-series analysis with autoregressive integrated moving average models. A total of 1071 subjects (18–22 years with ≥1 AD dispensation) were included in the study. The observed prevalence was lower than the predicted value for several weeks after the introduction of the lockdown. However, it was consistently higher than the predicted values from week 134. The maximum difference between observed and predicted values (25 subjects per 10,000 young adults) was found at week 170. Changes in AD use were observed in both genders and were more pronounced for selective serotonin reuptake inhibitors. In conclusion, the impact of the COVID-19 pandemic on the mental health of young adults is likely to be significant in the coming years, which may place a future burden on pharmaceutical public health and community health.

## 1. Introduction

Coronavirus disease 2019 (COVID-19), caused by severe acute respiratory syndrome coronavirus 2 (SARS-CoV-2), originated in Wuhan (China) in late 2019 and spread to all continents, becoming a global pandemic in just a few months [1]. Italy was one of the first and hardest-hit Western countries. The first cases were documented in the last days of February 2020, after which the incidence increased dramatically [2]. To prevent the spread of SARS-CoV-2 in the general population, the Italian government imposed a first nationwide quarantine (also called lockdown) from 9 March to 3 May (Phase 1) and progressively until 14 June (Phase 2) [3]. A second nationwide lockdown was imposed from mid-October to the end of the year due to a second wave of COVID-19 in the autumn [3]. During these periods, many restrictive measures were implemented, including the temporary closure of production activities and non-essential services, remote working, the adoption of online distance learning from primary schools to upper secondary schools and universities, the restriction of people’s movements except for reasons of necessity, and the restriction of physical interactions. These measures were modulated at the regional level according to both epidemiological metrics (e.g., the number of cases and their evolution over time) and the burden on the healthcare system.

The rapid spread of SARS-CoV-2 throughout the Italian territory together with the dangerousness of COVID-19 had a dramatic direct impact on healthcare and social systems [4,5,6]. In addition, changes associated with the context may have contributed to the increased public health burden [7,8,9]. However, in comparison with the effects of the COVID-19 pandemic, those related to the context appear to be more elusive. A deterioration in the mental health status of Italian citizens during the pandemic period has been described by many authors [10,11,12,13,14,15,16,17]. The data analyzed in most of these studies were collected through surveys and self-reported mental health status. Another methodological approach to document and monitor the mental health of the Italian population has been adopted by others. This approach is based on the quantification of psychotropic drug consumption, e.g., through the analysis of pharmaceutical administrative databases of the National Health System (NHS) or private companies to assess the utilization of antidepressants (ADs). Antonazzo et al. [18] found a decrease in the prevalence of AD users during the first lockdown, with a recovery after this period, reaching a higher level than predicted by the trend of AD dispensations before the lockdown in the Tuscany region. Pazzagli et al. [19] showed that AD consumption, described as defined daily doses (DDDs) per 1000 inhabitants per day [20], was higher in the first year of the COVID-19 pandemic when compared to the same month in 2015–2019 in the Friuli Venezia Giulia region. Marengoni et al. [21] found a decrease in AD consumption by older Italians (age over 65 years) when comparing DDDs per 1000 inhabitants between 2020 and 2019. A report by the Medicines Utilisation Monitoring Centre [22], led by the Italian Medicines Agency, documented an increasing trend in AD consumption (DDD per 1000 inhabitants per day) in Italy from 2013 to 2022, with no obvious change over the pandemic period. Finally, Marazzi et al. [23] studied the dispensations of ADs (DDD per month) by the totality of Italian pharmacies from 1 January 2019 to 31 December 2020. Compared to the pre-pandemic period, monthly purchases of ADs increased in March 2020 and decreased in the following 2 months. During the summer months, the differences in purchases decreased and did not increase again until November 2020. Therefore, contrasting results have been reported when studying the impact of the COVID-19 pandemic on AD utilization in Italy. The reasons for these discrepancies are not clear. Some methodological aspects (e.g., the length of the observation period, the metric adopted to quantify the AD use) may have masked changes of this outcome. In addition, certain subgroups of the general population may have needed medication to treat their psychological distress during the lockdown periods and/or after the reopening phase. In this respect, the association between younger age and lower levels of mental health during the COVID-19 pandemic has been documented. In particular, young adults have been found to experience high rates of loneliness, anxious and depressive symptoms, stress, and sleep disturbances [24,25,26,27,28,29,30,31,32,33,34,35,36]. Whether these changes were associated with a parallel change in AD consumption needs to be better understood.

The aim of this retrospective observational study was to estimate the impact of the COVID-19 pandemic on AD utilization by young adults in the ASL TO4 Regione Piemonte, a Local Health Authority (LHA) including both urban and rural areas, comprising 174 municipalities in the province of Turin and meeting the health needs of approximately 505,000 inhabitants (12% of the overall regional population). In particular, drug dispensing data were analyzed to estimate changes in the prevalence of AD users in young adults before and during the COVID-19 pandemic.

## 2. Materials and Methods

### 2.1. Data Source and Management

The data analyzed are derived from electronic health records of the ASL TO4 Regione Piemonte collected monthly for administrative purposes. These databases proved to be a feasible data source for observational retrospective studies [37,38,39]. These records describe medications dispensed (filled prescriptions) by local pharmacies and reimbursed by the Italian NHS. The data includes the patient’s date of birth and gender, dispensation date, number of dispensed packages, drug name, active ingredient, and Anatomical Therapeutic Chemical (ATC) code [20].

The study adhered to the regulations outlined in the General Data Protection Regulation-EU 2016/679 and data were handled in accordance with existing privacy legislation. Data were anonymized at the source and the authors did not have access to any personally identifiable information. The use of anonymized drug dispensing data for research purposes was approved by the Ethics Committee of the ASL TO4 Regione Piemonte on 21 November 2022.

### 2.2. Subjects and Study Drugs

Eligible subjects were young adults (age 18–22 years) with at least one dispensation for an AD (ATC code N06A). The index date was the date on which the first prescription of an AD was filled within the index period (from 1 January 2018 to 1 November 2021—200 weeks). Study drugs were identified according to their ATC code, and they belong to the classes of ADs reimbursed by the Italian NHS: non-selective monoamine reuptake inhibitors (N06AA), selective serotonin reuptake inhibitors (SSRIs, N06AB) and other ADs (N06AX).

### 2.3. Data Analysis

The overall prevalence of AD users was estimated by dividing the number of subjects with at least one prescription of an AD (N06A) during the index period by the number of inhabitants living in the ASL TO4 Regione Piemonte on 1 January of each corresponding calendar year and belonging to the age group considered as the reference population. The data were also analyzed by stratifying the study population according to the AD classes used (N06AA, N06AB, N06AX, or their combinations). The Clopper–Pearson method was used to estimate the 95% confidence interval (CI) for the prevalence values. In addition, the use of ADs was compared by sex and expressed as an odds ratio (OR). The Wald method was used to estimate the 95% CI for OR values, while the Fisher exact test (for which OR = 1) was used to determine *p* > χ^2^ (*p*-value).

The weekly prevalence of AD users was the study outcome adopted in this study, and it was determined as described by Antonazzo et al. [18] with minor modifications. First, sequences of the weekly prevalence of AD users were created by dividing the number of active treatment episodes in each week within the index period by the number of inhabitants living in the ASL TO4 Regione Piemonte on 1 January of each corresponding calendar year and belonging to the age group considered as the reference population. Active treatment episodes were established as previously described [38]. Briefly, a longitudinal dataset of medication supply was created for each subject from both the dispensation dates and the number of dispensed packages. The number of days of drug supply was calculated by dividing the total amount of dispensed active substance by its DDD. A continued treatment episode was considered if a new dispensation occurred within an appropriate grace period. Because a variety of medicinal products with different prescription durations were available on the Italian market for most of the study drugs, the length of the grace period was defined as 1.5 times the specific prescription duration. The Clopper–Pearson method was used to estimate the 95% CI for prevalence values. Then, the impact of the COVID-19 pandemic on the study outcome was assessed by performing an interrupted time-series analysis using autoregressive integrated moving average (ARIMA) models, as described by Schaffer et al. [40]. The index period was divided into two segments: pre-pandemic (from 1 January 2018 to 8 March 2020—114 weeks) and pandemic (from 9 March 2020 to 1 November 2021—86 weeks) periods, and the values predicted by our ARIMA models in absence of the COVID-19 pandemic (counterfactual) were compared with the observed values.

All analyses were performed using the R statistical and programming language (version 4.0.5; https://cran.r-project.org/, accessed on 4 July 2023). Several add-on packages and their dependencies were used to perform the analysis: lubridate, stringi, doBy, dplyr, tidyr, tidyverse, epiR, astsa, forecast, zoo, ggplot2 and epiR.

## 3. Results

### 3.1. Study Population

A total of 1071 subjects (60.3% females and 39.7% males) met the inclusion criteria and were included in the study. As shown in Figure 1a, the estimated overall prevalence was 112.9 (95% CI, 106.2–119.8) per 10,000 young adults. Selective serotonin reuptake inhibitors (N06AB) were the most commonly prescribed ADs (73.1% of overall subjects; 82.5 [76.9–88.5] per 10,000 young adults), followed by other ADs (N06AX, 21.8%; 24.6 [21.5–27.9] per 10,000 young adults) and non-selective monoamine reuptake inhibitors (N06AA, 16.6%; 18.8 [16.1–21.7] per 10,000 young adults). Few people (12.2%) were prescribed ADs belonging to two or three classes. The OR values for the comparison of female vs. male were greater than 1 (female favor) for 7 and statistically significant (*p* < 0.05) for 3 out of 8 categories (Figure 1b). In total, 14 ADs were prescribed. Sertraline and clomipramine were the most (20.1 [17.6–22.8] per 10,000 young adults) and the least (0.7 [0.3–1.3] per 10.000 young adults) commonly prescribed ADs, respectively (Figure 1c). The OR values for the comparison of female vs. male were greater than 1 (female favor) for 9 and statistically significant (*p* < 0.05) for 8 out of 14 drugs (Figure 1d).

### 3.2. Weekly Prevalence of AD Users

As shown in Figure 2, the weekly prevalence of AD users increased during the pre-pandemic period (from 2 April 2018 to 8 March 2020). Moreover, predicted and observed prevalence differed for several weeks during the pandemic period. In particular, the observed prevalence was lower than the predicted value for several weeks after the introduction of the restrictive measures. The maximum difference (6 subjects per 10,000 young adults, OR 0.95 [0.71–1.27]) was found in week 125 (25 May 2020). However, the observed prevalence was consistently higher than the predicted values from week 134 (27 July 2020). The maximum difference between observed and predicted values (25 subjects per 10,000 young adults, OR 1.24 [0.96–1.60]) was found in week 170 (25 April 2021).

### 3.3. Prevalence of N06AB Users Stratified by Gender

The weekly prevalence of SSRI users is shown in Figure 3a. As for all ADs, weekly prevalence values increased during the pre-pandemic period for both males and females. In addition, predicted and observed prevalence differed for several weeks during the pandemic period, showing a mixed pattern. A decrease in the observed prevalence compared with the predicted values was determined in the weeks following the introduction of the restrictive measures. Analyzing the data according to gender, the largest differences (−6 and −8 subjects per 10,000 young adults; OR 0.91 [0.64–1.29] and 0.91 [0.68–1.21], respectively) were found in week 129 (22 June 2020) and week 121 (27 April 2020), respectively. However, the observed prevalence was consistently higher than the predicted values from week 134 (27 July 2020) and week 126 (1 June 2020), respectively. The maximum difference between observed and predicted values (15 and 25 subjects per 10,000 young adults; OR 1.21 [0.88–1.65] and 1.23 [0.96–1.59], respectively) was found in week 190 (25 April 2021) and week 200 (1 November 2021), respectively.

### 3.4. Prevalence of N06AX Users Stratified by Gender

The weekly prevalence of other AD users is shown in Figure 3b. Prevalence values increased during the pre-pandemic period for both males and females, albeit with rates less pronounced than those observed for SSRIs. In addition, predicted and observed prevalence differed for several weeks during the pandemic period, with a mixed trend. In fact, in comparison with the predicted values, a decrease in the observed prevalence was determined in the weeks following the introduction of the restrictive measures. The largest differences (−5 and −4 subjects per 10,000; OR 0.64 [0.28–1.49] and OR 0.81 [0.43–1.54]) were found in week 130 (29 June 2020) and week 120 (20 April 2020), respectively. The observed prevalence was higher than the predicted values for most weeks and from week 139 (31 August 2020) and week 136 (10 August 2020), respectively. The maximum differences between observed and predicted values (10 and 8 subjects per 10,000 young adults; OR 1.71 [0.89–3.32] and 1.39 [0.81–2.38]) were determined in week 171 (12 April 2021) and week 180 (14 June 2021), respectively.

### 3.5. Prevalence of N06AA Users Stratified by Gender

The weekly prevalence of non-selective AD users is shown in Figure 3c. Weekly prevalence increased during the pre-pandemic period for both men and women but at less pronounced rates than for other classes of ADs. In addition, predicted and observed prevalence were comparable for several weeks during the pandemic period.

## 4. Discussion

As shown by our results, the COVID-19 pandemic was associated with a change in the use of ADs by young adults in the ASL TO4 Regione Piemonte. Our ARIMA models reveal a mixed pattern, with months in which the observed weekly prevalence was lower than expected, and vice versa. In particular, after the initial drop in the weekly prevalence in the first weeks after the introduction of the first nationwide lockdown, a strong upward trend in the weekly prevalence of AD users was determined. This change started in the summer of 2020 and was still present at the end of the index period. The significance of this change is highlighted by three aspects. First, predicted values for all ADs were below the lower limits of the 95% CI from week 161 (1 February 2021) and for most of the following weeks (see Figure 2). Second, there were consistent changes when the most commonly prescribed drugs for anxiety, depressive symptoms, stress, and sleep disorders were analyzed, namely SSRIs (N06AB), and, at least in part, other ADs (N06AX). Finally, both males and females experienced the same type of changes, albeit with different amplitudes.

Our results extend previous findings on the impact of COVID-19 on the use of ADs, both in Italy and in other countries. In particular, as mentioned in the introduction, previous studies have been focused on the use of ADs by the entire Italian population [22,23], the elderly [21], or residents of different regions of Italy [18,19]. Therefore, to the best of our knowledge, this is the first study to highlight the impact of the COVID-19 pandemic on the use of ADs in young Italian adults. In contrast, there are studies in the literature describing the impact of the COVID-19 pandemic on the use of ADs by young adults in other countries. For example, Leong et al. [41] studied the changes in psychotropic medication dispensation rates before and during the COVID-19 pandemic in the general population living in the province of Manitoba (Canada). The incidence and prevalence of AD use were lower in the second quarter of 2020 and higher in the last quarter of the same year than the expected trend. These changes were reported in several age groups, including young adults. Levaillant et al. [42] conducted a nationwide study to describe the use of psychotropic drugs in France during the first year of the COVID-19 pandemic compared to the previous five years. They found a very large increase in the weekly number of AD users among people aged 12–25 years during the pandemic period. Maguire et al. [43] analyzed the uptake of prescribed psychotropic medications by individuals in Northern Ireland during the lockdown period (March to October 2020) compared with previous patterns in the pre-pandemic period (January 2012 to February 2020). Apart from the increase in the number of individuals receiving ADs observed in March 2020, medication uptake remained within expected limits from April 2020 to October 2020. Slightly fewer people aged 18–24 years received ADs than expected in the early months of the pandemic, but uptake was as expected in all other age groups. Estrela et al. [44] studied the prescription trends of anxiolytics, sedatives, hypnotics, and ADs in Portugal. In contrast to the previous reports, they found a decreasing trend in the prescriptions of ADs in both adolescents (8–17 years old) and adults (18–64 years old). Bliddal et al. [45] examined the use of psychotropic medication among all Danish individuals aged 5–24 years from 1 January 2017 to 30 June 2022. Compared with expected results based on pre-pandemic trends, they found an increase in the incidence of AD use, particularly in the 12–24 age group. Therefore, collectively, these findings and our results indicate that COVID-19 pandemic had an impact on the use of ADs by young adults in many countries, albeit in different manner. The reasons for these differences are not clear. We can speculate that they may depend on methodological reasons. They could also be influenced by differences in access to medicines or the prescribing behavior of physicians in different contexts. Finally, it is worth mentioning that Piedmont was one of the Italian regions most affected during the COVID-19 pandemic, with the most severe restrictions, and this may have had a more negative impact on the mental health of young adults than elsewhere. Interestingly, in line with previously cited studies on the Italian population [18,19,20,21,22,23], we found no changes in AD use comparable to young adults in other age groups, supporting the hypothesis that this subgroup of individuals had needed medication to treat their psychological distress during the lockdown periods and/or after the reopening phase. This evidence supports the association between younger age and lower levels of mental health during the COVID-19 pandemic [24,25,26,27,28,29,30,31,32,33,34,35,36]. We can also speculate that the increase in the use of ADs by young adults in the ASL TO4 Regione Piemonte may be related to prescriptions for the treatment of post-COVID-19 depression. In this regard, as reported by Xie et al. [46], those who had COVID-19 had higher rates of mental health disorders and AD use compared to those who did not have COVID-19.

### Strengths and Limitations

This study linked individual-level dispensing data for the entire population of the ASL TO4 Regione Piemonte with area-level demographic datasets to allow for an accurate examination of AD use over a relatively long period of time. The use of individual-level data provided the opportunity to obtain accurate weekly prevalence rates, overcoming the limitations associated with the use of population volume data. In addition, using ARIMA modelling, this study was able to incorporate long-term trends (over approximately the last 2 years) to predict expected trends for the pandemic period. This adjustment for pre-existing conditions allowed for a demonstration of the impact of the COVID-19 pandemic on the use of ADs. Applying ARIMA models to time-series health data has the advantage of allowing for autocorrelation, which is known to be present in trends and seasonality, both of which are known to be present in drug use [40].

Some limitations of the study should be mentioned. First, the LHA database records treatments provided and reimbursed by the Italian NHS, while non-reimbursed treatments are not recorded. In addition, as with other data collected primarily for administrative purpose, they may not reflect the actual use of AD by patients. In order to interpret the derived prevalence results correctly, these aspects must be taken into account. For example, our prevalence values appear to be lower than those estimated by the Medicines Utilisation Monitoring Centre for a similar age group [22]. However, these calculations were based on different data sources. Despite these limitations, it is unlikely that the observed changes in prevalence values do not reflect real changes in AD use. In support of this hypothesis, certain aspects that typically characterize AD use, such as greater use by women than men and greater use of SSRIs than other AD classes, are clearly evident in our results. Second, the diagnosis of mental health disorders by both specialist and general practitioners were not available. Nevertheless, we are reassured that the patterns of AD use are consistent with the known literature on the distribution of mental illness within the population. Third, although we used ATC classification to group the medications, it should be remembered that some of the medications belonging to the class of ADs can also be prescribed to treat conditions other than mental health disorders. A significant difference between males and females was found for the use of amitriptyline (see Figure 1d). We can speculate that this finding is not only related to the use of this drug for the treatment of mental disorders but rather may reflect its use for other conditions for which it is indicated. These include migraine (prophylaxis in adults), the prevalence of which in young adults is higher in women than in men [47]. Moreover, this approach may mask some specific trends for a particular drug. However, the resulting misclassification is unlikely to change during the pandemic. Fourth, the use of other psychotropic drugs commonly prescribed to treat anxious and depressive symptoms, stress, and sleep disturbances (e.g., benzodiazepines) was not taken into account, nor were other age groups. In both cases, the main reason was the lower accuracy of the prescription data compared with the data analyzed, which could have led to more uncertain conclusions. Finally, ARIMA models assume that the dataset is normally distributed and homoscedastic, meaning it has a constant variance, which may not be true for the time-series data analyzed. In addition, like other predictive models, uncertainty for forecasts grows rapidly with longer periods of time into the future, so caution should be taken when using ARIMA models for long-term forecasting. ARIMA models are not suitable for multivariate time-series data and cannot capture the interactions and dependencies between different variables, such as the effects of external factors on the time series. For all these reasons, we have refrained from carrying out a formal statistical analysis to compare the predicted and observed values of the weekly prevalence rates.

## 5. Conclusions

The use of dispensing data as a proxy for mental health without information on diagnosis is a recognized limitation. In fact, drug dispensing data could underestimate the actual need for psychiatric therapies due to the low recourse to adequate treatment in the event of a mental disorder. Our results were derived from a relatively small study population living in an Italian region severely affected by the COVID-19 pandemic. These aspects may make it difficult to generalize the conclusions. Nevertheless, together with previous epidemiological research, the results of this study suggest that Italian young adults in the ASL TO4 Regione Piemonte experienced a deterioration of their mental health status during the COVID-19 pandemic period. The underlying causal factors are unknown, but the restrictive measures implemented during the lockdowns, abrupt changes in family routines, social isolation and even the direct effects of COVID-19 could be some of the factors of potential importance. Regardless of the underlying cause, the impact of the COVID-19 pandemic on the mental health of young adults is likely to be significant in the coming years, which may place a future burden on pharmaceutical public health and community health. Future studies could enhance our understanding of the phenomenon by incorporating broader demographics, including a comparative analysis with regions not impacted by the pandemic, and using a wider range of data sources. In addition, in the coming years, it will be important to design and implement programs and interventions that are able to anticipate and address the mental distress experienced by young adults during and after the COVID-19 pandemic; at the same time, it will be useful to improve the use of ADs in this age group.

## Figures and Tables

**Figure 1 pharmacy-12-00021-f001:**
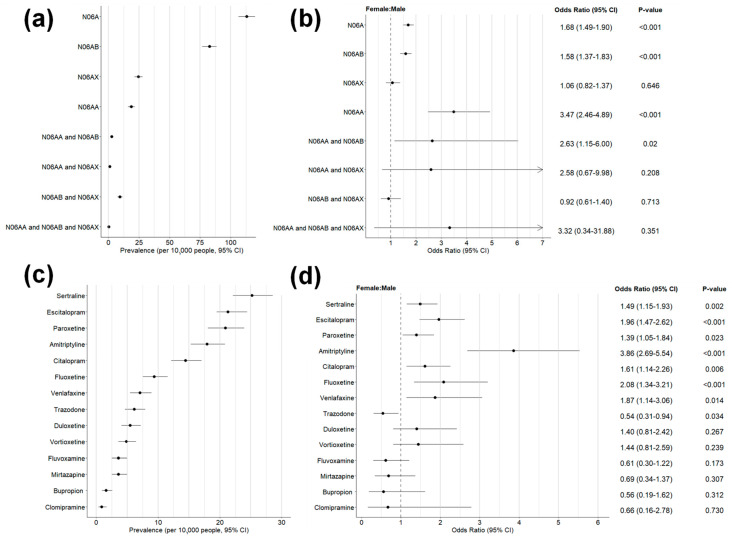
Use of ADs in the study population (*n* = 1071 subjects). The overall prevalence of AD users (95% CI) was calculated and stratified according to the ATC classification system: panel (**a**) all ADs (N06A), non-selective monoamine reuptake inhibitors (N06AA), SSRIs (N06AB) and other ADs (N06AX); panel (**b**) AD use (classified as drug classes) compared by genders and expressed as odds ratio (95% CI); panel (**c**) single drugs; panel (**d**) AD use (classified as single drugs) compared by genders and expressed as odds ratio (95% CI).

**Figure 2 pharmacy-12-00021-f002:**
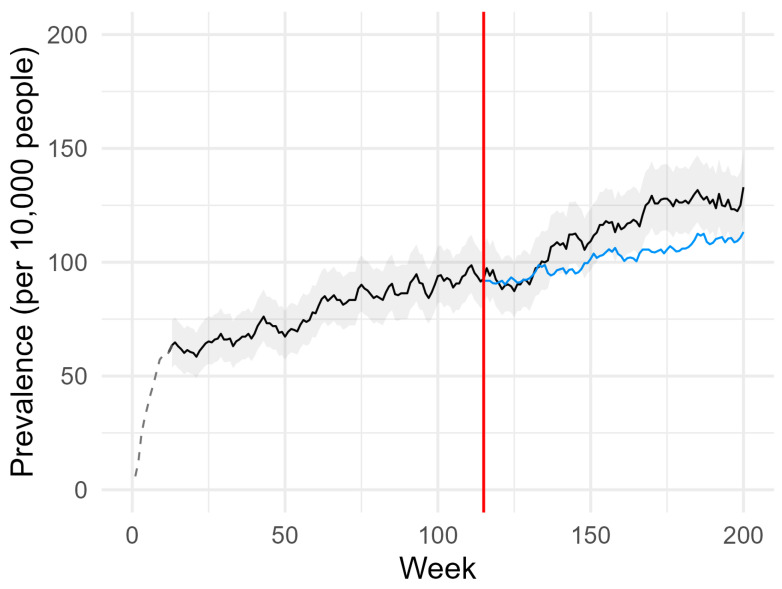
Weekly prevalence of AD users. The number of active treatment episodes (all ADs, ATC code N06A) in each week of the index period was divided by the number of inhabitants living in the ASL TO4 Regione Piemonte on 1 January of each corresponding calendar year and belonging to the age group considered as the reference population. The black (longer line) and blue (shorter line) lines represent the observed and predicted values, respectively. The grey area is the 95% CI, and the dotted line is the unmodelled data (12 weeks). The red line represents the start of the lockdown.

**Figure 3 pharmacy-12-00021-f003:**
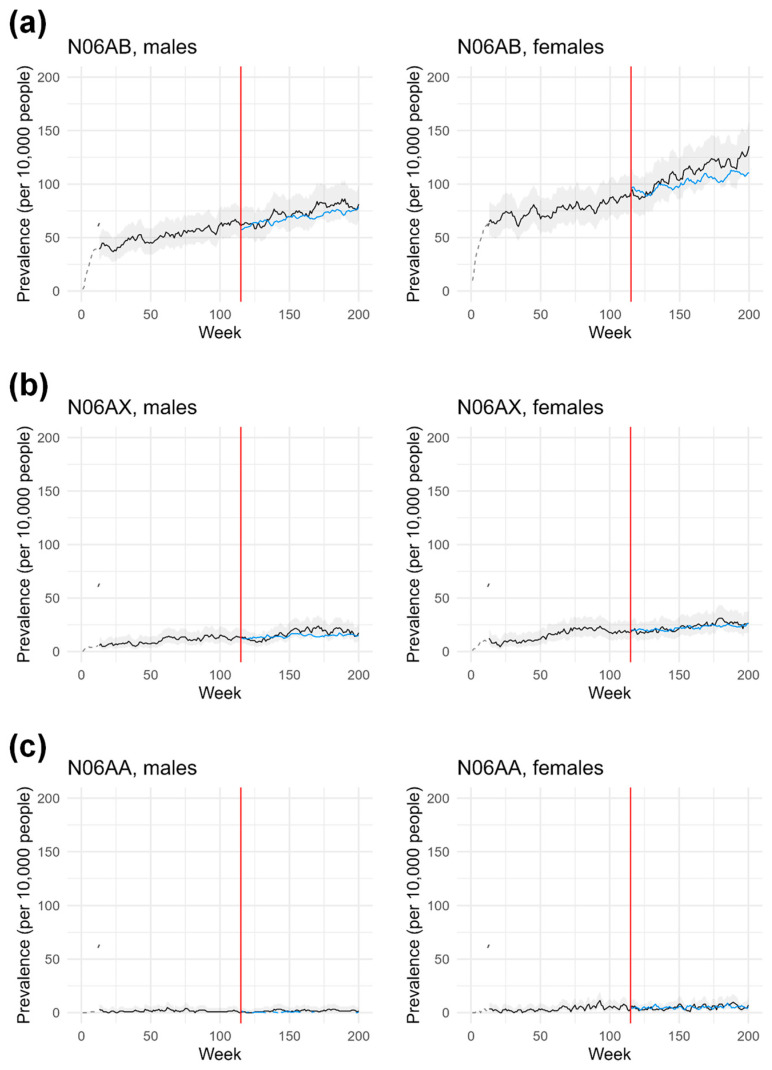
Weekly prevalence of AD users stratified by drug class and sex. The number of active treatment episodes (ATC code N06AB, panel (**a**); N06AX, panel (**b**); N06AA, panel (**c**)), in each week of the index period was divided by the number of inhabitants living in the ASL TO4 Regione Piemonte on 1 January of each corresponding calendar year and belonging to the age group considered as the reference population. The black (longer line) and blue (shorter line) lines represent the observed and predicted values, respectively. The grey area is the 95% CI, and the dotted line is the unmodelled data (12 weeks). The red line represents the start of the lockdown.

## Data Availability

All data relevant to this study are included in the manuscript.

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
