# Peer review of "Impact of the COVID-19 Pandemic on the Use of Antidepressants by Young Adults in the ASL TO4 Regione Piemonte (Italy)"

_pharmacy, 2024, doi:10.3390/pharmacy12010021_

Round 1
Reviewer 1 Report
Comments and Suggestions for Authors
The manuscript presents a timely and relevant research study examining the change in antidepressant usage among young adults during the COVID-19 pandemic. The research is significant in understanding how a global health crisis affects mental health treatment patterns in a specific demographic.
- The primary question addressed is the impact of the COVID-19 pandemic on antidepressant usage among young adults in the ASL TO4 Regione Piemonte. This focus is crucial as it delves into the intersection of public health crises and mental health treatment patterns.
- The topic is both original and relevant. The study fills a significant gap in current literature by focusing on the change in antidepressant usage during an unprecedented global health crisis. This is particularly important given the rising concerns about mental health during the pandemic.
- The authors should acknowledge that while their methodology using interrupted time series with ARIMA models was suitable for their study's scope, it had limitations. They could note that the study's focus on a specific region and demographic might limit the generalizability of the findings. Additionally, they might suggest that future studies could enhance understanding by incorporating broader demographics, including a comparative analysis with regions not impacted by the pandemic, and using a wider range of data sources. This acknowledgment would demonstrate an awareness of the methodology's limitations while underscoring the relevance and strengths of the current study within its context.
- In the discussion section, the authors should consider expanding their commentary on the implications of their findings, particularly in relation to broader public health strategies and mental health support during crises. They could also delve deeper into the nuances of gender differences in antidepressant use, exploring societal and healthcare factors contributing to these trends. Furthermore, discussing potential future research directions, based on the limitations and findings of their study, would provide valuable context and suggest ways to build on their work. This approach would enhance the depth and applicability of their discussion.
- The references used are appropriate, lending credibility and context to the research. They support the study's findings and are well-integrated into the overall narrative of the manuscript.
Author Response
Response to Reviewer 1
We are grateful to the Reviewer 1 for his/her efforts in reviewing our manuscript and for his/her comments and suggestions.
The manuscript presents a timely and relevant research study examining the change in antidepressant usage among young adults during the COVID-19 pandemic. The research is significant in understanding how a global health crisis affects mental health treatment patterns in a specific demographic.
The primary question addressed is the impact of the COVID-19 pandemic on antidepressant usage among young adults in the ASL TO4 Regione Piemonte. This focus is crucial as it delves into the intersection of public health crises and mental health treatment patterns.
The topic is both original and relevant. The study fills a significant gap in current literature by focusing on the change in antidepressant usage during an unprecedented global health crisis. This is particularly important given the rising concerns about mental health during the pandemic.
The authors should acknowledge that while their methodology using interrupted time series with ARIMA models was suitable for their study's scope, it had limitations.
The Discussion section was revised accordingly to this suggestion (see page 9, lines 335-343).
They could note that the study's focus on a specific region and demographic might limit the generalizability of the findings. Additionally, they might suggest that future studies could enhance understanding by incorporating broader demographics, including a comparative analysis with regions not impacted by the pandemic, and using a wider range of data sources. This acknowledgment would demonstrate an awareness of the methodology's limitations while underscoring the relevance and strengths of the current study within its context.
The Conclusion section has been revised accordingly to this suggestion (see page 9, lines 359-361).
In the discussion section, the authors should consider expanding their commentary on the implications of their findings, particularly in relation to broader public health strategies and mental health support during crises. They could also delve deeper into the nuances of gender differences in antidepressant use, exploring societal and healthcare factors contributing to these trends. Furthermore, discussing potential future research directions, based on the limitations and findings of their study, would provide valuable context and suggest ways to build on their work. This approach would enhance the depth and applicability of their discussion.
The Manuscript (several parts) has been revised accordingly.
The references used are appropriate, lending credibility and context to the research. They support the study's findings and are well-integrated into the overall narrative of the manuscript.

Reviewer 2 Report
Comments and Suggestions for Authors
The current research article manuscript is an interesting study on the impact of the COVID-19 pandemic on antidepressant drug use by young adults in a specific region of the country of Italy. The results are carefully presented and discussed, albeit some discussion could still be improved, as well as other relevant aspects. Hence, I advise for the following alterations to be made before acceptance for publication:
- In the introduction section, some long sentences lack appropriate referencing, this should be corrected;
- Statistical analysis of the collected data should be performed, in order to better support result interpretation;
- An attempt should be made to compare antidepressant use between genders, and also different ages; this should be done for all antidepressant drugs in general, then specific classes, then specific drugs; this analysis should then be compared with already described associations between gender or age with antidepressants (already existing studies);
- For better reader visualization, Tables 1 and 2 should be transformed into figures (graphical representations, for example using bar graphs);
- The obtained data should be better compared with other already existing studies, namely in other regions of Italy, in Europe, and worldwide;
- An abbreviation list is missing and should be added.
Author Response
Response to Reviewer 2
We are grateful to the Reviewer 2 for his/her efforts in reviewing our manuscript and for his/her comments and suggestions.
The current research article manuscript is an interesting study on the impact of the COVID-19 pandemic on antidepressant drug use by young adults in a specific region of the country of Italy. The results are carefully presented and discussed, albeit some discussion could still be improved, as well as other relevant aspects. Hence, I advise for the following alterations to be made before acceptance for publication:
In the introduction section, some long sentences lack appropriate referencing, this should be corrected;
The Introduction section has been revised accordingly to this suggestion (see pages 1-2). Moreover, seven new references have been added to the bibliography.
Statistical analysis of the collected data should be performed, in order to better support result interpretation;
Statistical analysis of the collected data has been performed. In particular, the study population has been described in more detail (see below), and the differences between predicted and observed weekly prevalence were reported as both absolute and relative values (Odds Ratio, 95% CI). The Methods and Results sections have been revised consistently.
An attempt should be made to compare antidepressant use between genders, and also different ages; this should be done for all antidepressant drugs in general, then specific classes, then specific drugs; this analysis should then be compared with already described associations between gender or age with antidepressants (already existing studies);
A formal inferential analysis based on a statistical test for the comparison between predicted and observed values has not been performed. The predicted values (ARIMA models) are affected by uncertainty due to the limitations of these models (see page 9, lines 335-343). Thus, cautions should be taken using ARIMA models for long-term forecasting. However, the significance of these results is demonstrated by the consistency of the results. In contrast, an analysis based on statistical hypothesis testing was performed to compare the observed data. In particular, prevalence values were calculated for the study population stratified by categories (N06A, N06AA, N06AB, N06AX, their combinations, single drugs) as well as between males and females. No comparison was made with other age groups. As suggested by our preliminary analyses, the use of ADs in the ASL TO4 Regione Piemonte was largely age-dependent. Younger age groups differed significantly from older ones, in agreement with previous studies (see manuscript for details). However, our results suggest that changes in the use of ADs occurred during the transition from the pre-pandemic to the pandemic period, but with different signs and entities for different age groups. As a complex and non-obvious scenario emerges from these preliminary analyses, we have decided to limit the comparison to genders rather than age. This will hopefully be the subject of a future publication. The manuscript has been revised consistently.
For better reader visualization, Tables 1 and 2 should be transformed into figures (graphical representations, for example using bar graphs);
Table 1 and 2 have been transformed into Figure 1.
The obtained data should be better compared with other already existing studies, namely in other regions of Italy, in Europe, and worldwide;
Previous studies in other Italian regions, Europe and worldwide on this topic have been described in more detail in both the Introduction and Discussion sections. In particular, the similarities and differences with our data have been highlighted.
An abbreviation list is missing and should be added.
An abbreviation list has been added.

Reviewer 3 Report
Comments and Suggestions for Authors
The manuscript pharmacy-2775056 entitled Impact of COVID-19 Pandemic on the Use of Antidepressants by Young Adults in the ASL TO4 Regione Piemonte (Italy) by Lucrezia Greta Armando and coworkers aimed to estimate the impact of the COVID-19 pandemic on AD use by young adults in the ASL TO4 Regione Piemonte (Italy). A total of 1,071 subjects (18-22 years with ≥1 AD dispensation) were included in the study. The observed prevalence was lower than the predicted value for several weeks after the introduction of the lockdown. However, it was consistently higher than the predicted values from week 134. The maximum difference between observed and predicted values (25 subjects per 10,000 young adults) was found at week 170. Changes in AD use were observed in both genders and were more pronounced for selective serotonin reuptake inhibitors. It is likely that the impact of the COVID-19 pandemic on the mental health of young adults would be significant in the future.
The work is scientifically robust and methodology is appropriate.
Figures and tables are informative
A minor linguistic revision is recommended.
Comments on the Quality of English LanguageA minor linguistic revision is recommended.
Author Response
Response to Reviwer 3
We are grateful to the Reviewer 3 for his/her efforts in reviewing our manuscript and for his/her comments and suggestions.
The manuscript pharmacy-2775056 entitled Impact of COVID-19 Pandemic on the Use of Antidepressants by Young Adults in the ASL TO4 Regione Piemonte (Italy) by Lucrezia Greta Armando and coworkers aimed to estimate the impact of the COVID-19 pandemic on AD use by young adults in the ASL TO4 Regione Piemonte (Italy). A total of 1,071 subjects (18-22 years with ≥1 AD dispensation) were included in the study. The observed prevalence was lower than the predicted value for several weeks after the introduction of the lockdown. However, it was consistently higher than the predicted values from week 134. The maximum difference between observed and predicted values (25 subjects per 10,000 young adults) was found at week 170. Changes in AD use were observed in both genders and were more pronounced for selective serotonin reuptake inhibitors. It is likely that the impact of the COVID-19 pandemic on the mental health of young adults would be significant in the future.
The work is scientifically robust and methodology is appropriate.
Figures and tables are informative
A minor linguistic revision is recommended.
English was revised.

Reviewer 4 Report
Comments and Suggestions for Authors
Thank you for your submission. It is quite interesting, despite there being numerous similar publications from Italy and worldwide.
There should be some interpretation and discussion around the actual absolute prevalence rates of antidepressant use. The values (appearing to be only about 1.2% of the population in the relevant age group) seem to be very low relative to published data internationally. Why is this? Perhaps there has been long-standing under-use in the region?
Language such as “the COVID-19 pandemic had an impact on the use of ADs by young adults in the ASL TO4 Regione Piemonte” should be re-worded to avoid any implication of a definitive cause and effect relationship e.g. “the COVID-19 pandemic was associated with use of ADs by young adults in the ASL TO4 Regione Piemonte”.
The English is generally good, but I am unsure what this means on page 1: “the restriction of people’s except for reasons of necessity”.
Comments on the Quality of English LanguageThe English is generally good, but I am unsure what this means on page 1: “the restriction of people’s except for reasons of necessity”.
Author Response
Response to Reviewer 4
We are grateful to the Reviewer 3 for his/her efforts in reviewing our manuscript and for his/her comments and suggestions.
Thank you for your submission. It is quite interesting, despite there being numerous similar publications from Italy and worldwide.
There should be some interpretation and discussion around the actual absolute prevalence rates of antidepressant use. The values (appearing to be only about 1.2% of the population in the relevant age group) seem to be very low relative to published data internationally. Why is this? Perhaps there has been long-standing under-use in the region?
According to the latest report published by the Italian Medicines Utilisation Monitoring Centre, in a similar age group (15-24 years), the prevalence of AD users in 2022 was less than 2.5%. Our estimates, both for overall prevalence and for weekly prevalence, are lower (~1%). The reasons for this discrepancy are unclear, but some hypotheses can be made. Firstly, it is well known that AD use in Italy shows some variability between regions, with regions of high use (e.g., Tuscany) and low use (e.g., Campania). Piedmont is in an intermediate position. We can therefore assume that the differences observed reflect the territorial context. Secondly, the administrative data used by the Italian Medicines Utilisation Monitoring Centre include both reimbursed and non-reimbursed prescriptions, whereas those used in our study are those of drugs reimbursed by the NHS. Therefore, it is possible that our results underestimate the actual use of AD by young adults. Nevertheless, this type of data is routinely used for research purposes and has been shown to provide a satisfactory description of patterns of use and their changes over time. This point has been highlighted in the manuscript (see page 8, lines 312-323).
Language such as “the COVID-19 pandemic had an impact on the use of ADs by young adults in the ASL TO4 Regione Piemonte” should be re-worded to avoid any implication of a definitive cause and effect relationship e.g. “the COVID-19 pandemic was associated with use of ADs by young adults in the ASL TO4 Regione Piemonte”.
The English is generally good, but I am unsure what this means on page 1: “the restriction of people’s except for reasons of necessity”.
English was revised.

Round 2
Reviewer 1 Report
Comments and Suggestions for Authors
I would like to thank the Authors for all improvements. I have no further comments.
Reviewer 3 Report
Comments and Suggestions for Authors
The manuscript was properly improved.
The authors answered to all the points